# Uncertainty-sensitive Learning and Planning with Ensembles

## Abstract

We propose a reinforcement learning framework for discrete environments in which an agent makes both strategic and tactical decisions. The former manifests itself through the use of value function, while the latter is powered by a tree search planner. These tools complement each other. The planning module performs a local *what-if* analysis, which allows to avoid tactical pitfalls and boost backups of the value function. The value function, being global in nature, compensates for inherent locality of the planner. In order to further solidify this synergy, we introduce an exploration mechanism with two distinctive components: uncertainty modelling and risk measurement. To model the uncertainty we use value function ensembles, and to reflect risk we use propose several functionals that summarize the implied by the ensemble. We show that our method performs well on hard exploration environments: Deep-sea, toy Montezuma's Revenge, and Sokoban. In all the cases, we obtain speed-up in learning and boost in performance.

## 1 Introduction

The model-free and model-based approaches to reinforcement learning (RL) have a complementary set of strengths and weaknesses. While the former offers good asymptotic performance, it suffers from poor sample complexity. In contrast, the latter usually needs significantly less training samples, but often fails to achieve state-of-the-art results on complex tasks (which is primarily attributed to models' imperfections). The interplay of model-based and model-free approaches in RL has received a lot of research attention. This led, for example, to strong AI systems like Silver et al. (2017; 2018) or more recently to Lowrey et al. (2018), which is closely related to our work.

When dealing with challenging new RL domains it is helpful to develop tools for addressing strategic and tactical decision-making. These two perspectives complement each other: strategic perspective is global, static, and often imprecise, while tactical perspective is local, dynamic and exact. We argue that value function can be considered as an example of the former, while a planner as an example of the latter. Indeed, value function approximators provide noisy estimates (imprecise) of values (static) to every state (global). Conversely, planning provides optimal control, which starting from a given state (local) generates actions (dynamic) that are temporally coherent and result in a better-executed trajectories (exact). It is only reasonable to combine the two into one system. This is achieved by plugging the value function into the planner, to provide guided heuristics in the local search, shorten the search horizon, and make the search computationally efficient. For complex problems, the above setup has to be supplemented by an exploration strategy. Such a strategy might be based on modeling uncertainty of the value function approximation (e.g., using dropout sampling, variational inference, distributional RL, bootstrap ensemble, etc.). The uncertainty is quantified by a risk measure, which is then utilized by a planner to guide exploration.

Consider a situation of an agent with limited memory and computational resources, being dropped into a complex and diverse environment. This setup excludes any hope for brute force strategy and implies that a perfect planner is out of reach. In this paper, we consider Sokoban, a classic logical puzzle known for its combinatorial complexity and a recent benchmark for RL, ChainEnv, a seemingly impossible task also known as a 'hay in a needle-

stack' problem (see Osband et al. (2018)), and Toy Montezuma's Revenge, environment notoriously known for its exploration difficulty (see Guo et al. (2019)). Importantly, we focus on sparse reward variants of the aforementioned environments (see Section 3). Since of our method involves planning, an agent has to have an access to the world model. In this paper we assume that it is the perfect model. The reason is that challenges presented by the selected environments are the dominating factor driving difficulty of the task (in fact, answering the question of whether a Sokoban level is solvable is NP-hard, see e.g. Dor & Zwick (1999)). It is also in part due to this fact that a significant body of work dealing with difficult environments makes a similar assumption (see e.g. Silver et al. (2017; 2018), Orseau et al. (2018b)).

The main contribution of this work is showing how recent progress in AI can be brought together to improve planning, value function learning, and exploration, in a way that together they form robust algorithms for solving challenging reinforcement learning environments. In particular:

1. For uncertainty modeling, we assume the point of view Osband et al. (2018), which uses ensembles to approximate posterior distribution.
2. In the spirit of Lowrey et al. (2018), we incorporate risk measures to guide exploration.
3. For the planner, we base on AlphaZero Monte-Carlo Tree Search (see Silver et al. (2017)), which we enhance with the possibility of exploiting the graph structure of environments and the properties of the optimal solutions.
4. In the value function training protocol, we introduce several improvements, including a version of self imitation learning mechanism (Oh et al. (2018a)) and hindsight (Andrychowicz et al. (2017)).

The rest of the paper is organized as follows. In the next subsection we provide an overview of related work. In Section 2 we present and discuss our method in details. This is followed by a section with experiments and passing to conclusions. We provide code to our work `https://github.com/learningandplanningICLR/learningandplanning` and a dedicated website `https://sites.google.com/view/learn-and-plan-with-ensembles` with more details and movies.

## 1.1 Related work

The ideas of mixing model-based and model-free learning were perhaps first stated explicitly in Sutton (1990). Many approaches followed. More recently, in the groundbreaking series of papers Silver et al. (2017), Silver et al. (2018) culminating in AlphaZero, the authors have developed an elaborate system that plans and performs model-free training to master the game of Go (and others). Similar ideas were also studied in Anthony et al. (2017).

Perhaps, the work which is closest to ours is Lowrey et al. (2018). It is argued in the paper that an agent with limited computational resources in a complex environment needs both to plan and learn from the incoming stream of experience. Importantly, the value function in Lowrey et al. (2018) is modeled by an ensemble of value functions. The risk measure used to combine them is given by the 'log-sum-exp formula' (Lowrey et al., 2018, 6). The authors show experimentally that this approach leads to improvements in various continuous tasks, including humanoid training. In our work, we deal with a discrete setting, which enforces a different planning module (here MCTS-based). Moreover, we treat more diverse class of risk measures (see Section 2).

Constructing neural network models that would incorporate uncertainty in a principled Bayesian way has proven to be challenging and remains and open problem. A promising new results using ensembles include Osband et al. (2018; 2017), Lakshminarayanan et al. (2017). Ensembles of models were also successfully used to improve model-based RL training, see Kurutach et al. (2018), Chua et al. (2018), and the references therein. We note that unprincipled ensemble methods are willingly used by RL practitioners. For example, in a recent competition Kidzinski et al. (2019) aimed to train an agent able to use a prosthetic leg, four top-ten solutions used some sort of ensembles.

Another work similar to ours is Guo et al. (2014), in which the authors use MCTS in the role of an 'expert' from which a neural policy is learnt using the dagger algorithm, Ross & Bagnell (2014). The basic difference is that Guo et al. (2014) uses a classical MCTS without value function nor ensembles.

Many works aim to build planning and learning into neural network architectures, see e.g. Oh et al. (2017) or Farquhar et al. (2017). Kaiser et al. (2019), a recent work on model-based Atari, has shown the possibility of sample efficient reinforcement learning with an explicit visual model. Gu et al. (2016) uses model-based at the beginning phase of training and model-free methods in 'fine-tuning'. Furthermore, there is a body of work that attempts to learn the planning module, see Pascanu et al. (2017), Racanière et al. (2017), and Guez et al. (2019).

Finally, our paper is related to research focusing on study of exploration. Fundamental results in this area concern the multi-arm bandits problem, see Lattimore & Szepesvári (2018) and the references therein. Methods developed in this area have been successfully applied in planning algorithms, see Kocsis et al. (2006) and Silver et al. (2017; 2018). Furthermore, a measure with a loading on variance (defined in Section 2) is related to UCB-V algorithm developed in Audibert et al. (2007). Another set of methods have been developed in an attempt to solve notoriously hard Montezuma's Revenge, see for example Ecoffet et al. (2019) or Guo et al. (2019).

In our work we use also hindsight, see Andrychowicz et al. (2017). Its primarily motivation is to enrich the learning signal and train generalized value function. However, in certain situation it can be considered as an exploration algorithm.

## 2 Method description

In this section, we describe our method: the planning and exploration components, as well as the training protocol. Algorithm 1 shows how the components are brought together in the training loop. The pseudo-code for `planner.run_episode` is listed in Algorithm 2, and the remainder of pseudo-code can be found in Appendix A.

---

**Algorithm 1** Learning and planning with ensembles

---

**Require:** Environment `env`, Model `model`
 1: Initialize parameters of value function ensemble $\theta = (\theta_1, \ldots, \theta_K)$, $\mathbf{V}_\theta = (V_{\theta_1}, \ldots, V_{\theta_K})$
 2: Initialize `replay_buffer`
 3: **repeat**
 4:     $s \leftarrow$ `env.reset`()
 5:     $\text{episode, solved} \leftarrow$ `planner.run_episode`$(s; \texttt{model}, \mathbf{V}_\theta)$          $\triangleright$ see Algorithm 2
 6:     $\text{values} \leftarrow$ `evaluate_episode`(episode)          $\triangleright$ see Algorithm 8
 7:     Optionally calculate a `mask`          $\triangleright$ see Appendix C
 8:     `replay_buffer.add`(episode, values, solved, mask)          $\triangleright$ see Algorithm 9
 9:     $B \leftarrow$ `replay_buffer.batch`()          $\triangleright$ $B = \{(s_b, v_b, m_b)\}$, see Algorithm 10
10:     Update $\mathbf{V}_\theta$ by one step of gradient descent          $\triangleright$ e.g. RMSProp

$$\nabla_{\theta_i} \left( \frac{1}{|B|} \sum_{(s,v,m) \in B} m_i \big( V_{\theta_i}(s) - v \big)^2 + \zeta \theta_i^2 \right), \qquad \text{for} \qquad i \in \{1, \ldots, K\}$$

11: **until** convergence

---

For the planer component we develop an MCTS-inspired algorithm.[1] It is novel in two aspects: it utilizes a risk-sensitive policy (for tree traversal and action choice) intended

---

[1]MCTS is a family of algorithms that iteratively build a search tree, alternating between the following stages: tree traversal, leaf expansion and evaluation, and backpropagation, see Browne et al. (2012) for a survey on the topic.

to guide exploration, and it includes methods exploiting the graph structure and enables avoiding cycles.[2]

The loop avoidance is achieved in two ways: by backpropagation of some fixed negative value through the in-tree path ending with a leaf having no unvisited neightbours (see the pseudo-code in Appendix A related to `dead_ends`), and during tree traversal, when the agent encouraged to avoid actions leading to previously visited states on the path (see pseudo-code related to $\texttt{penalty}_p$ in Appendix A). To reinforce this effect, the agent is also encouraged to avoid actions leading to previously visited states on the episode level (see Algorithm 2, where $\texttt{penalty}_e$).

These enhancements combined, make it possible to learn even in sparse rewards scenarios, which is experimentally demonstrated in Section 3 (all used environments have sparse rewards). We note that Algorithm 1 is not MCTS specific, other planners could be used as well.

The logic of our MCTS is laid out in Algorithm 2. We assume that each node of the search tree, say `n`, stores a visit count `n.count`, accumulated value `n.value`, accumulated reward `n.reward(a)`, and its children are denoted by `n.child(a)` for each action $a \in \mathcal{A}$. We define

$$\widehat{Q}_\theta(\mathtt{n}, a) = \mathtt{n.reward}(a) + \gamma \mathtt{n.child}(a).\mathtt{value},$$

where $\gamma > 0$ is a discount rate.

---

**Algorithm 2** `planner.run_episode()`

**Require:**   `max_episode_len`
              `mcts_passes`
              $\texttt{penalty}_e$ ▷ Episode penalty
              $\mathbf{V}_\theta$        ▷ Value function
**Input:**     `s`         ▷ Starting state
1: $\texttt{episode} \leftarrow \emptyset$
2: **for** $\texttt{step} = 1$ to `max_episode_len` **do**
3:     $\texttt{root} \leftarrow \texttt{s}$
4:     $\texttt{root.value} \leftarrow \texttt{root.value} - \texttt{penalty}_e$
5:     **for** 1 to `num_mcts_passes` **do**
6:         $\texttt{path}, \texttt{leaf} \leftarrow \texttt{traversal(root)}$
7:         $\texttt{value} \leftarrow \texttt{expand\_leaf(leaf;} \mathbf{V}_\theta)$
8:         $\texttt{backpropagate(value, path)}$
9:     $\texttt{a} \leftarrow \texttt{choose\_action(root, \{root\})}$
10:    $\texttt{s, r, done} \leftarrow \texttt{model.step(a)}$
11:    $\texttt{episode.append((root, a, r))}$
12:    **if** `done` **then break**
13: **return** `episode, done`

---

**Algorithm 3** `choose_action()`

**Require:** `avoid_loops`              ▷ Bool
**Input:** `n, seen`
1: **if** `avoid_loops` **then**
2:     $A \leftarrow \{a \in \mathcal{A}(\mathtt{n}) \colon \mathtt{n.child(a)} \notin \mathtt{seen}\}$
3:     **if** $A = \emptyset$ **then** ▷ Terminal or dead-end
4:         **return** `None`
5: **else**
6:     $A \leftarrow \mathcal{A}(\mathtt{n})$
7: Choose action according to equation 1

$$a^* \leftarrow \underset{a \in A}{\arg\max}\, \mathbb{E}_{\theta \sim \Theta}\left[\phi_a(\widehat{\mathbf{Q}}_\theta(\mathtt{n}))\right]$$

8: **return** $a^*$

---

Our proposed risk-sensitive exploration method is implemented at the level of MCTS traversal tree, see step 7 in Algorithm 3. Its key defining elements are uncertainty modeling and risk measurement. A risk-sensitive tree traversal policy is defined as follows:

$$a^*(\mathtt{n}) := \arg\max_a \mathbb{E}_{\theta \sim \Theta}\left[\phi_a(\widehat{\mathbf{Q}}_\theta(\mathtt{n}))\right], \quad \widehat{\mathbf{Q}}_\theta(\mathtt{n}) := \left(\widehat{Q}_\theta(\mathtt{n}, a') \colon a' \in \mathcal{A}\right), \tag{1}$$

where $\mathcal{A}$ is the action space, $\phi_a \colon \mathbb{R}^{|\mathcal{A}|} \to \mathbb{R}$ is a risk measure, and $\widehat{Q}_\theta$ is an estimator of the $Q$-function. The posterior distribution $\Theta$ models uncertainty in the value estimation.

We implement the posterior $\Theta$ using ensembles, namely $\mathbb{E}_{\theta \sim \Theta}[f(\theta)]$ is approximated by $\frac{1}{K}\sum_{i=1}^K f(\theta_i)$, where $K$ is the size of an ensemble. This approximation stems from Osband et al. (2018, Lemma 3), which shows that for Bayesian regression in a Gaussian linear

---

[2]This is useful for a broad class of environments (including the ones we used in the experimental part) for which the optimal trajectory does not have cycles.

model, generating samples from posterior distribution is equivalent to solving a suitable optimization problem, see Appendix C. In our case we use an ensemble of value functions. In some cases, we "sub-sample" from $\Theta$, which is inspired by Osband et al. (2016) and the classical Thomson sampling (see Appendix C for details).

Using risk measure is inspired by Lowrey et al. (2018), who used $\phi_a(x) = e^{\kappa x_a}$ for $x \in \mathbb{R}^{|\mathcal{A}|}$ and $\kappa > 0$. In this paper we consider the following choices of $\phi_a$:

- A measure with a loading on variance, $\phi_a(x) = x_a + \kappa x_a^2$, $\kappa > 0$. This includes second moments and can be easily generalized to include variance, standard deviation and exploration bonuses.

- A relative majority vote (also known as plurality vote) measure,

$$\phi_a(x) = 1\left(\arg\max_{a'} x_{a'} = a\right). \tag{2}$$

  Contrary to the other cases, $\phi_a$ defined in equation 2 depends not only on marginal values of its input, but the whole input (i.e. the estimator vector $\widehat{\mathbf{Q}}_\theta$). It leads to a rule resembling optimal Bayes classifier form, i.e. the one which chooses $a$ minimizing $\mathbb{P}(a_\theta^*(s) \neq a)$.

The intuitions behind the aforementioned choices of $\phi_a$'s are as follows. One can think of $\phi_a(x) = e^{\kappa x_a}$ used in Lowrey et al. (2018) as a measure capturing all moments of value ensemble. For small values of $\kappa \approx 0$, it behaves like the measure with a loading on variance (via the Tylor expansion). The mean approximates the value of a given action, while the variance quantifies the epistemic uncertainty. Taking weighted sum of these terms (as we do in "mean+var" experiments, see Table 3) has the aim of balancing exploitation and exploration. It is also related to UCB-V algorithm, see Audibert et al. (2007). Voting, on the other hand, is a well established approach when combining ensembles, see e.g. Breiman (1996) or Rokach (2010). A relative majority vote, in particular, is simple and it can lead to good performance, see e.g. Osband et al. (2016). In the context of planning and RL, it has several interesting properties. In particular, the distribution of votes across ensembles encodes the uncertainty related to the optimal action in a given state. High uncertainty may result in stochastic movement (caused by a uniform tie breaking), and consequently lead to higher exploration. On the other hand, low uncertainty may result in an exploitative behaviour of an agent and, as a result, lower exploration. Additionally, voting may improve decision making of an agent in the states, where some action can be dangerous (e.g. lead to irreversible states).

We conclude this section by describing the part of Algorithm 1 related to value functions update. The episodes are collected and stored in a replay buffer. The replay buffer performs some addition bookkeeping by storing, for each transition, information whether it comes from a solved episode or not (in the case of environments that provide such information, like Sokoban). This mechanism allows sampling batches with a fixed ratio of solved to unsolved transitions (see Algorithm 8 in Appendix A). Such a method resembles self-imitation techniques, see e.g. Oh et al. (2018b). The value functions are trained to minimize $l_2$ distance from target values sampled from the buffer. The target values can be computed in two modes: `bootstrap`, which utilizes the values accumulated during the MCTS phase, and `factual`, corresponding to discounted rewards in an episode (see Algorithm 8 in Appendix A). Additionally, some experiments use masks which, analogously to Osband et al. (2018), form a mechanism of assigning a transition to a value function (see Appendix C). Finally, we implement ensembles as a set of neural networks (we also experiment with multi-headed architectures sharing some lower layers). We use both architectures with or without random priors, for details see Appendix B and Osband et al. (2018).

## 3 EXPERIMENTS

In this section, we provide experimental evidence to show that using ensembles and risk measures is useful. We chose three environments, Deep-sea, toy Montezuma's Revenge, and

Sokoban. In all the cases we work with single reward versions of the these environments i.e. the agent's is rewarded only upon success completion of its task.

We use an MCTS planner with the number of passes equal to 10 (see line 4 of Algorithm 2). We consider this number to be rather small for MCTS-like planning methods. Interestingly, we observed that such a relatively weak planner is sufficient to obtain a well-preforming algorithm.

We utilize various neural network architectures, see Appendix B. We measure uncertainty using either variance or standard deviation except for the case of Sokoban with randomly-generated boards, where voting was used, see equation 2.

Configuration of the experiments is summarized in Table 3 in Appendix C.

### 3.1 Deep-sea

Deep-sea environment was introduced in (Osband et al., 2018, Section 4) and later included in Osband et al. (2019) as a benchmark for exploration. The agent is located in the upper-left corner (position $(0,0)$) on a $N \times N$ grid, $N \in \mathbb{N}$. In each timestep, its $y$-coordinate is

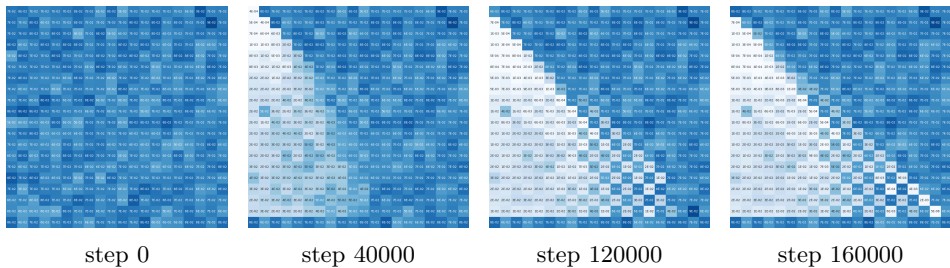

| step 0 | step 40000 | step 120000 | step 160000 |

Figure 1: The heatmaps of standard deviations of ensemble values in the Deep-sea environment. High values are marked in blue and low in white. At the beginning of training (left picture) the standard deviation is high for all states. Gradually it is decreased in the states that have been explored. Finally (the right) the reward state is found. Note that the upper-right part of the board is unreachable.

increased, while $x$ is controlled. The agent issues actions in $\{-1, 1\}$. These are translated to *step left* or *step right* (increasing $x$ by $-1$ or $+1$, respectively, as long as $x \geq 0$; otherwise $x$ remains unchanged) according to a prescribed action mask (not to be confused with transition masks in Algorithm 1). For each *step right* the agent is punished with $0.01/N$. After $N$ steps, the game ends, and the agent receives reward $+1$ if and only if it reaches position $(N, N)$. The action mask mentioned above is randomized at each field at the beginning of training (and kept fixed).

Such a game is purposely constructed so that naive random exploration schemes fail already for small $N$'s. Indeed, a random agent has chance $(1/2)^N$ of reaching the goal even if we disregard misleading rewards for *step right*. In Figure 1, one can observe how the exploration progresses of our method. In Figure 2, one can see a comparison of non-ensemble models, ensemble models with Thomson sampling (see Appendix C) but without uncertainty bonus ($\kappa = 0$) and our final ensemble model with uncertainty bonus $\kappa = 50$. We conclude that both using sub-sampling and ensembles is essential to achieve good exploration (for details see also Appendix C).

### 3.2 Toy Montezuma's Revenge

Toy Montezuma's Revenge is a navigation maze-like environment. It was introduced in Roderick et al. (2018) to evaluate ideas of using higher-level abstractions in long-horizon problems. While its visual layer is greatly reduced version of the actual Montezuma's Revenge Atari game, it retains much of the original's exploration problems. This makes it a useful test environment for exploration algorithms, see e.g. Guo et al. (2019). In our experiments

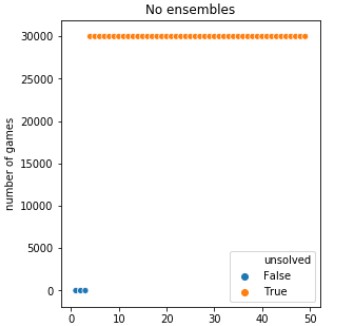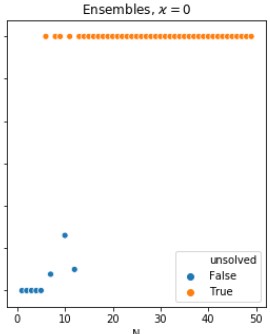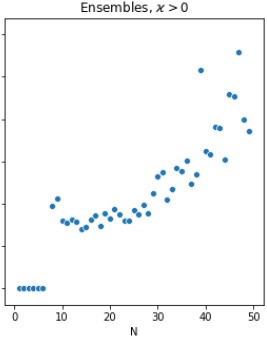

Figure 2: Comparison of number of episodes needed to solve the deep-sea environment with given grid size $N$. Orange dots marks trials which were unable to solve problem in 30000 episodes. Large problem instances ($N > 20$) are solved only when exploration bonus is used (right-most plot, $\kappa = 50$)

we work with with the biggest map with 24 rooms, see Figure 3.[3] In order to concentrate on the evaluation of exploration we chose to work with sparse rewards. The agent gets reward 1 only if it reaches the treasure room, otherwise the episode is terminated after 300 steps.

It is expected that any simple exploration technique would fail in this case (we provide some baselines in Table 1). Guo et al. (2019) benchmarks PPO, PPO with self imitation learning (PPO+SIL), PPO with count based exploration bonus (PPP+EXP) and their new technique (DTSIL). Only DTSIL is consistently able to solve 24 room challenge, with PPO+EXP occasionally reaching this goal. Our method based on ensembles solves this exploration challenge even in a harder, sparse reward case.[4] The results are summarized in Table 1. We have three setups:

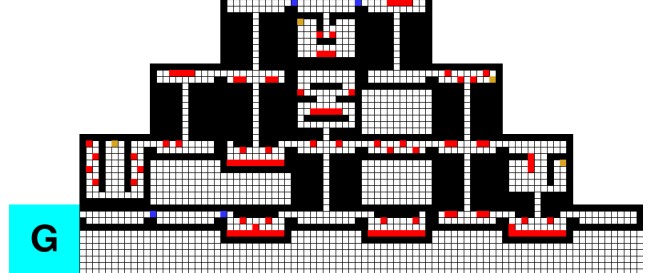

Figure 3: The biggest Montezuma's Revenge map, consisting of 24 room. The goal is to reach the room marked with $G$. The agent needs to avoid traps (marked in red) and pass through doors (marked in blue). Doors are open using keys (marked in yellow).

*no-ensemble*; *ensemble, no std*; *ensemble, std*. In the first case, we train using Algorithm 1 with a single neural-network. In the second case, for each episode we sub-sample 10 members of ensemble of 20 to be used and the MCTS is guided by their mean. In the final third case, we follow the same protocol but we add to the mean the standard deviation. In our experiments we observe that *no-ensemble* in 30 out of 43 cases does not leave behind the first room. The setup without explicit exploration bonus, *ensemble, no std*, perform only slightly better. Finally, we observe that *ensemble, std* behaves very well.

Further experimental details and the network architecture are presented in Appendix B and C.

### 3.3 SOKOBAN

Sokoban is an environment known for its combinatorial complexity. The agent's goal is to push all boxes (marked as yellow, crossed squares) to the designed spots (marked a

---

[3]We use a slightly modified code from `https://github.com/chrisgrimm/deep_abstract_q_network`

[4]DTSIL builds on the intermediate partial solutions, which are ranked according to their reward, thus we suspect it would fail in the sparse reward case.

| Setup | win-ratio (no. seeds) | av. visited rooms |
|---|---|---|
| *no-ensemble* | 0 (43) | 4.8 |
| *ensemble, no std* | 2 (40) | 5.8 |
| *ensemble, std* | 35 (37) | 17.1 |

Table 1: Result for toy Montezuma's Revenge. We report the number of successful runs and the number of seeds of network initialization. We also show the average number of visited rooms, which is a proxy for the learning progress.

square with red dot in the middle), see Figure 4. Apart from the navigational factor, the difficulty of this game is greatly increased by the fact that some actions are irreversible.

A canonical example of such an action is pushing a box into a corner, though there are multiple less obvious cases. Formally, this difficulty manifests itself in the fact that deciding whether a level of Sokoban is solvable or not, is NP-hard, see e.g. Dor & Zwick (1999). Due to these challenges the game has been considered as a testbed for reinforcement learning and planning methods.

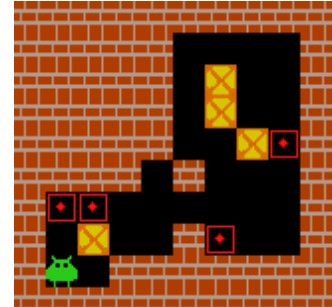

Operationally, to generate Sokoban levels we use an automated procedure proposed by Racanière et al. (2017). Some RL approaches to solving Sokoban (e.g. Racanière et al. (2017), Guez et al. (2019)) assume additional reward signal in the game, e.g. for pushing a box into a designed spot. In this work we use *sparse setting*, that is the agent is rewarded only when all the boxes are put into place.

Figure 4: Example $(10, 10)$ Sokoban board with 4 boxes. Boxes (yellow) are to be pushed by agent (green) to designed spots (red). The optimal solution in this level has 37 steps.

It is interesting to note, that Sokoban offers two exploration problems: single-level-centric, where a level-specific exploration is needed, and multi-level-centric, where a 'meta-exploration' strategy is required, which works in a level-agnostic manner or can quickly adapt. As a result, we conducted three lines of experiments using ensembles: *a) learning to solve randomly generated boards(dubbed as multiple-boards Sokoban), b) learning to solve a single board (dubbed as single-boards Sokoban) , c) transfer and learnable ensembles.*

In our experiment we use Sokoban with board of size $(10, 10)$ and 4 boxes. We use the limit of 200 steps in the experiment *a)* and 100 in the remaining ones.

**Multiple-board Sokoban, learning to solve randomly generated boards** In this experiment we measure the ability of our approach to solve randomly generated Sokoban boards. We measure the performance of the agent by computing the win rate on last 1000 games. In this experiment we use an ensemble value function using relative majority voting as formalized in equation 2.

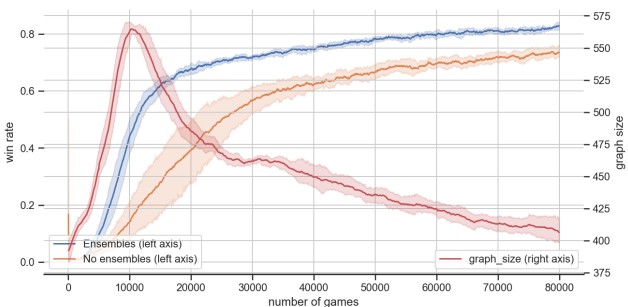

Figure 5: Learning curve (left axis) and the size of explored graph (right axis) of Sokoban states. The shape of the latter plot may show a gradual switch from exploration to exploitation.

Relative majority voting takes into account the uncertainty of ensembles when it comes to the final outcome, not only the uncertainty in assessment of particular action. After 80000 games, our method reaches 85% win rate, compared to 76% of an agent not using ensembles, see Figure 5. As a measure of exploration

| Architecture | Random | Trans. 1 | Trans. 2 | Trans. 3 |
|:---:|:---:|:---:|:---:|:---:|
| 5-layers | 0.7% | 4.9% | 7.1% | 8.5 % |
| 4-layers | 0.7% | 4.3% | 5.6% | 7.3% |

Table 2: Results of transfer experiments. We test transfers from one value function (Trans 1) and transfer from ensembles of 2 and 3 value functions (Trans 2 and Trans 3). In the later two cases the aggregation of values is learned. The results are averaged over 20 seeds.

we also present the size of the game graph explored during episodes (red curve in Figure 5). It shows an interesting effect, which can be interpreted as a transition from exploration to exploitation approximately at a 40% win-rate mark or, equivalently, 10000 games.

**Single-board Sokoban. Learning to solve a single board** In this experiment we measure the extent in which our methods can plan and learn on single boards. We note that this setting differs substantially from the one in the previous paragraph. In the multiple-boards Sokoban there is a possibility of generalization from easier to harder scenarios (e.g. randomly initialized network solves 0.7% of boards). Such a transfer is obviously not possible in the single-board case studied here, however there are many attempts on the same board, which enables the agent to explore the board's state space.

For the single-board Sokoban we used experimental settings similar to the one in Section 3.2, see also details in Appendix C. We observe that the setup with ensembles solves 73% compared to 50% the standard training without ensembles. The latter might seem surprisingly good, taking into account the sparse reward. This follows by the loop avoidance described in Section 2. If during planning the agent finds itself in a situation from which it cannot find a novel state (i.e. encounters `dead-end` in Algorithm 6) a negative value, set to $-2$ in our experiments, is backpropagated in Algorithm 5 to already seen vertices. We speculate that this introduces a form of implicit exploration.

In the singe-board Sokoban case we performed also experiments on $(8, 8)$ boards with 4 boxes obtaining success rate of 94% compared to 64% on the standard non-ensemble settings.

**Transfer and learnable ensembles** Generating any new board can be seen as a cost dimension along with sample complexity. This quite naturally happens in meta-learning problems. We tested how value functions learned on small number of boards perform on new previously unseen ones. We used the following protocol, we trained value function on fixed number of 10 games. To ease the training we used relabeling akin to Andrychowicz et al. (2017)[5]. We evaluate these models on other boards. It turns out that they are typically quite weak, arguably it is not very surprising as solving 10 boards does not give much chance to infer 'the general' solutions. In the second phase we use ensembles of the models. More precisely, we calculate the values of $n = 2, 3$ models and aggregate it using a small neural networks with one fully connected hidden layer. This network is learnable and trained using the standard setup. We observe that the quality of such ensemble increases with the number of components as summarized in Table 2. We observed high variability of the results over seeds, this is to be expected as board in Sokoban significantly vary in difficulty. We also observe that maximal results for transfer increase with the number of value function, being approximately 10%, 11% and 12%. This further supports the claim that ensembling may lead to improved performance. In 5-layer experiment we use a network with 5 hidden cnn layers, see details in Appendix B, we compare this with an analogous 4 layers network. In the latter case, we obtain weaker result. We speculate this might be due to the fact that smaller architecture is easier to overfit.

---

[5]More precisely, for a failing trajectory we choose a random time-step and shift the target spots so that they match the current location of boxes. We note that although this operation requires the knowledge of the game mechanics (i.e. its perfect model) it is used only in this phase

## Conclusions and further work

In this paper, we introduced a reinforcement learning method that blends planning, learning, and risk-sensitive approach to exploration. We verified experimentally that such a setup is useful in solving hard exploration problems, i.e. problems characterized by sparse rewards and long episodes (e.g. spanning even hundreds of steps).

We believe that this opens promising future research directions. There are multiple ensemble design choices, and we tested only a selected few. Additionally, there are more ways for combining the results of ensembles and it would be interesting to see if one, relatively general, method can be found. Such a result would be a step towards deep Bayesian learning.

In our work, we used MCTS with a perfect model of deterministic environments. It would be interesting to consider problems requiring the use of learned, imperfect models. This is a more demanding task, and it is intriguing whether the approach to uncertainty developed here could be useful in dealing with model imperfections. Equally important research direction would be related to solving stochastic environments. This might be considerably more difficult as such a task requires disentangling epistemic (studied in this work) and aleatoric (coming from the environment) uncertainties.

We focused our attention on the Monte-Carlo Tree Search, but there is a priori no reason why some other planning should not yield better results. In some initial experiments we obtained promising, but yet inconclusive, result using the Levin tree search (see Orseau et al. (2018a)). Another tempting direction is training both value function and a policy, akin to methods of Silver et al. (2017).

It would also be interesting to isolate the proposed exploration method from the planner and see how it fares when coupled with model-free algorithms. This fits along the lines of some recent research directions, see e.g. Agarwal et al. (2019).

Going further, we speculate that it may be possible to use the methods developed for Sokoban in the meta-learning and continual learning grounds problems. Perhaps measures of uncertainty can be used to enable a learning system to adapt to a changing environment. In an archetypical case, this might be obtained by choosing from ensemble a model (a skill) which is useful at the moment and understanding situations that such a model is not yet present.

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

## A  Algorithm

In this section we detail the building blocks of Algorithm 1. As mentioned in Section 2, the key elements of MCTS (see Algorithm 2) are tree traversal, leaf expansion, and backpropagate, shown in Algorithm 4, Algorithm 6 and Algorithm 5, respectively. Algorithm 7 shows the update of a tree node. In particular, Algorithm 6 shows how the value function enters the picture, by evaluating all of the leaf's children (line 9). Algorithm 4 and Algorithm 5 both use variable $\texttt{penalty}_p$. This is a penalty corresponding to entering the same states during tree traversal (hence is operates on the planner level) and the change is applied during traversal, and undone during backpropagation.

---

**Algorithm 4** `traversal()`

---

**Require:** $\texttt{penalty}_p$ $\qquad\qquad$ ▷ Planner penalty
**Input:** `root`

1: $\texttt{n} \leftarrow \texttt{root}$
2: $\texttt{path} \leftarrow \emptyset$
3: **while** `n` is not a leaf **do**
4: $\qquad \texttt{n.value} \leftarrow \texttt{n.value} - \texttt{penalty}_p$
5: $\qquad a \leftarrow \texttt{choose\_action(n, path)}$
6: $\qquad$ **if** $a$ is None **then** $\qquad$ ▷ Dead-end
7: $\qquad\qquad$ **break**
8: $\qquad \texttt{path.append((n},a))$
9: $\qquad \texttt{n} \leftarrow \texttt{n.child}(a)$
10: **return** $\texttt{path, n}$ $\qquad\qquad$ ▷ $\texttt{n} \notin \texttt{path}$

---

**Algorithm 5** `backpropagate()`

---

**Require:** $\texttt{penalty}_p$, $\gamma$
**Input:** `v, path`

1: **for** $(n,a)$ in `reversed(path)` **do**
2: $\qquad \texttt{n.value} \leftarrow \texttt{n.value} + \texttt{penalty}_p$
3: $\qquad \texttt{v} \leftarrow \texttt{n.reward} + \gamma\texttt{v}$
4: $\qquad \texttt{update(n, v)}$

---

**Algorithm 6** `expand_leaf()`

---

**Require:** $\texttt{dead\_end\_value}, \widehat{\mathbf{V}}_\theta$
**Input:** `leaf`

1: **if** `leaf` is terminal **then**
2: $\qquad \texttt{update(leaf, 0.)}$
3: $\qquad$ **return** $0.$
4: **else if** `leaf` is a dead-end **then**
5: $\qquad \texttt{update(leaf, dead\_end\_value)}$
6: $\qquad$ **return** $\texttt{dead\_end\_value}$
7: **else**
8: $\qquad$ **for** $a \in \mathcal{A}(\texttt{leaf})$ **do**
9: $\qquad\qquad \texttt{v} \leftarrow \widehat{\mathbf{V}}_\theta(\texttt{leaf.child}(a))$
10: $\qquad\qquad \texttt{leaf.child}(a)\texttt{.value} \leftarrow \texttt{v}$
11: **return** $\texttt{leaf.value}$

---

**Algorithm 7** `update()`

---

**Input:** `n, value`

1: $\texttt{n.value} \leftarrow \texttt{n.value} + \texttt{value}$
2: $\texttt{n.count} \leftarrow \texttt{n.count} + 1$

---

The following blocks of code are related to the training setup. Algorithm 8 is responsible for computing an appropriated value for each element of the episode. There are two available modes: `"bootstrap"`, which utilizes the values accumulated during the MCTS phase, and `"factual"`, which represents the sum of discounted rewards in the episode. In the `"bootstrap"` mode we undo the penalty applied during the episode generation stage (line 4 in the Algorithm 2). The inner details of replay buffer, in particular Hindsight and self-imitation, is given in Algorithm 9 and Algorithm 10. Hindsight refers to any method that processes episode, and potentially overrides states values (see lines 1-2 in Algorithm 9). Algorithm 10 shows the way a batch is generated. First, for each transition it is determined whether it should be sampled from a solved or unsolved episode (according to a fixed ratio). Second, a game is sampled from a population determined in the previous step, with probability proportional to the game length. Finally, given a game a transition is chosen uniformly.

**Algorithm 8** `evaluate_episode()`

---

**Require:** mode          ▷ String
         $\text{penalty}_e$ ▷ Episode penalty

**Input:**    episode     ▷ $\{(s_t, a_t, r_t)\}$
         solved           ▷ Bool
         $\gamma$             ▷ Discount rate

---

1: $T \leftarrow$ `len(epsiode)`
2: $\text{values} \leftarrow (0, \ldots, 0) \in \mathbb{R}^T$
3: **if** mode=`"bootstrap"` **then**
4:     $\text{rewards} \leftarrow \{s_t.\text{value} + \text{penalty}_e\}$
5: **else if** mode=`"factual"` **then**
6:     $\text{rewards} \leftarrow [0, \ldots, 0, \text{solved}]$
7: **for** $t = T - 1$ to $0$ **do**
8:     $\text{values}_{t-1} \leftarrow \gamma \text{values}_t + \text{rewards}_t$
9: **return** values

---

**Algorithm 9** `replay_buffer.add()`

---

**Require:** $\mathcal{D}$       ▷ Replay Buffer
         $H$    ▷ Hindsight mapping

**Input:**    episode ▷ $\{(s_t, a_t, r_t, w_t)\}$
         value         ▷ $\{v_t\}$
         solved        ▷ Bool
         mask      ▷ Binary vector

---

1: **for** $t = 0$ to $T - 1$ **do**     ▷ Hindsight
2:     $v_t \leftarrow H(\text{episode})_t$
3: $\mathcal{D} \leftarrow \mathcal{D} \cup (\{(s_t, a_t, v_t)\}, \text{solved}, \text{mask})$

---

**Algorithm 10** `replay_buffer.batch()`

---

**Require:** $\mathcal{D}$      ▷ Replay Buffer
         size       ▷ Batch size
         ratio ▷ Solved/unsolved

---

1: Initialize $B \leftarrow \emptyset$
2: **for** $b = 1$ to size **do**
3:     **if** $b \times \text{ratio}\%1 = 0$ **then**
4:        $\text{select} \leftarrow$ `False`
5:     **else**
6:        $\text{select} \leftarrow$ `True`
7:     $D' \leftarrow \{d \in \mathcal{D} : d.\text{solved} = \text{select}\}$
8:     Sample $d \in \mathcal{D}'$ with

$$\text{prob}(d) \propto \text{len}(d.\text{episode})$$

9:     Sample $(s, v, m)$ uniformly from $d$
10:     $B \leftarrow B \cup \{(s, v, m)\}$
11: **return** $B$

---

# B   ARCHITECTURES

**Deep-sea**   For Deep-see environment we encode state as one hot vector of size $N^2$, and learn simple linear transformation for value estimation.

**Toy Montezuma Revenge**   Observation is represented as tuple containing current room location, agent position within the room and status of all keys and doors on the board. To estimate value we use fully-connected neural networks with two hidden layers consisting 50 neurons each.

**Single-board Sokoban**   Observation has a shape $(10, 10, 7)$ where first two coordinates are spatial and the third coordinate one-hot encodes type of a state (e.g. box, target, agent, wall). To estimate value we flatten the observation and apply fully-connected neural networks with two hidden layers consisting 50 neurons each.

**Multiple-boards Sokoban**   Here we use the same observation type as in the single-board problem. Each value function network is composed of five $3 \times 3$ convolutional layers with with stride 1, followed by two fully connected layers with 128 units and 1 unit, respectively.

# C   TRAINING DETAILS

**Randomized priors**   In Osband et al. (2018, Lemma 3) showed that for Bayesian linear regression setting with Gaussian prior and noise model, generating samples from posterior distribution is equivalent to solving an appropriate optimization problem. To be exact,

suppose $\mathcal{D} = \{(x_i, y_i)\}$ is the dataset, $f_\theta(x) = x^T\theta$ is the regression function, $\epsilon_i$ is a Gaussian noise, $\tilde{\theta}$ comes from a Gaussian prior, and $\tilde{y}_i = y_i + \epsilon_i$. Then the solution of the following problem

$$\arg\min_\theta ||\tilde{y}_i - (f_{\tilde{\theta}} + f_\theta)(x_i)||_2^2 + \zeta||\theta||_2^2, \tag{3}$$

for some $\zeta > 0$, is a sample from the posterior $\theta|\mathcal{D}$. Consequently, Osband et al. (2018) propose to use equation 3 as a training objective and include randomized prior in the value function approximator. This objective is matched with the one used in Algorithm 1, and some of our experiments include randomized priors.

**Masks**   An important decision for training, is how to assign transitions to the particular elements of value functions ensemble. This is implemented using masks, see Algorithm 1. Suppose a batch $B$ is considered in an update step (see lines 9-10 of Algorithm 1) a let $\mathtt{t} = (s, v, m) \in B$ be a transition. A mask $m \in \{0, 1\}^{\{1,\dots,K\}}$ has the following interpretation: $m_i = 1$ if and only if the transition $\mathtt{t}$ is used to train $V_{\theta_i}$, $i = 1, \dots, K$.

We experimented with the following versions of masking:

- dynamic masks: masks are generated anew whenever transition is sampled from replay buffer,
- static masks: each transition is assigned a fixed mask generated when added to the replay buffer.

In the cases of dynamic masks, each batch was split equally among the elements of ensemble (for this we kept the batch size to be the multiplicity of the number of ensembles). The static masks were inspired by the Bootstrapped DQN, see (Osband et al., 2016, Appendix B), where it is a core idea. We experimented with applying a different mask to each transition according to the Bernoulli distribution, or to assign the same masks for all transitions in the same trajectory.

We found it useful to use static masks in the Deep-sea, Toy Montezuma's Revenge and Single-board Sokoban experiments, and dynamic masks in Multiple-board Sokoban experiments.

**Ensembles sampling**   Recall that equation 1 involves calculating the expected value, $\mathbb{E}_{\theta \sim \Theta}$, with respect to the posterior distribution $\Theta$. In some experiments we instead sub-sample from $\Theta$. Such an approach follows Osband et al. (2016), which in turns is inspired by the classical Thomson sampling (see discussion (Osband et al., 2016, Section 4) and the original Thompson (1933)). To be concrete, for a given a risk measure $\phi_a$ equation 1 reduces to

$$a^*(\mathtt{n}) := \arg\max_a \sum_{i \in \mathcal{E}} \left[\phi_a(\widehat{\mathbf{Q}}_{\theta_i}(\mathtt{n}))\right], \quad \widehat{\mathbf{Q}}_\theta(\mathtt{n}) := \left(\widehat{Q}_\theta(\mathtt{n}, a'): a' \in \mathcal{A}\right), \tag{4}$$

where $\mathcal{E} = \{1, \dots, K\}$. Sub-sampling, with a fixed parameter $\ell$, is equivalent to computing equation 4, with $\mathcal{E}$ taken as a random subset of $\{1, \dots, K\}$ of cardinality $\ell$.

**Hiperparameters**   Table 3, below, summarizes the parameters of the training used in the Deep-see, Toy Montezuma's Revenge, Single-board Sokoban and Multiple-boards Sokoban experiments presented in Section 3.

| Parameter | Deep-sea | Toy MR | Single-b. Sok. | Multiple-b. Sok. |
|---|---|---|---|---|
| Number of MCTS passes[1] | 10 | 10 | 10 | 10 |
| Ensemble size $K$[2] | 20 | 20 | 20 | 3 |
| Ensemble sub-sampling $\ell$[3] | 10 | 10 | 10 | no |
| Risk measure[4] | mean+std, | mean+std | mean+std | voting |
| $\kappa$[5] | 50 | 3 | 9 | n/a |
| VF target[6] | bootstrap | bootstrap | bootstrap | factual_discount |
| Discounting $\gamma$[7] | 0.99 | 0.99 | 0.99 | 0.99 |
| Randomized prior[8] | no | no | no | yes |
| Optimizer[9] | RMSProp | RMSProp | RMSProp | RMSProp |
| Learning rate[10] | $2.5e-4$ | $2.5e-4$ | $2.5e-4$ | $2.5e-4$ |
| Batch size[11] | 32 | 32 | 32 | 32 |
| Mask[12] | static | static | static | dynamic |

[1] `num_mcts_passes` in the MCTS algorithm, see Algorithm 2
[2] the number of value functions in ensemble in Algorithm 1
[3] the parameter of ensemble sub-sampling
[4] *mean+std* stands for mean $+ \kappa \cdot$ std, where mean is the mean of the ensemble predictions and std is its standard deviation. *voting* stands for using as given by equation 2
[5] see footnote 4
[6] see description in Section 2 and Algorithm 8
[7] as used in Algorithm 8
[8] see Section 2 and Appendix C
[9] optimizer used in step 10 of Algorithm 1
[10] optimizer's learning rate
[11] cardinality of batch $B$ in line 9 of Algorithm 1
[12] line 10 of Algorithm 1

Table 3: Default values of hyperparameters used in our experiments.

