# OpenReview forum: "Uncertainty - sensitive learning and planning with ensembles"
_ICLR.cc/2020/Conference — Reject_

### Official Review · AnonReviewer1 · 2019-10-23
**Official Blind Review #1**

**Rating:** 6

**Review:**

The paper proposes an approach blending model-based, model-free methods and utilizing risk-sensitivity information in ensembles as part of the value estimation and exploration process. The exploration is based on risk- sensitivity measures such as moments and relative majority vote. There is a lot of work currently trying to marry the model free with model based approaches for integrated planning and learning as the authors have mentioned in the related work section of the paper and also called out similar methods and techniques. The authors have provided evidence via experiments in three environments and shown good results of using this blended approach. Code is also provided for others to further carry out explorations in this research area.


**Experience Assessment:**

I have read many papers in this area.

**Review Assessment: Checking Correctness Of Derivations And Theory:**

I assessed the sensibility of the derivations and theory.

**Review Assessment: Checking Correctness Of Experiments:**

I assessed the sensibility of the experiments.

**Review Assessment: Thoroughness In Paper Reading:**

I made a quick assessment of this paper.

---

> ### Author Response · Authors · 2019-11-15
> **Answer to AnonReviewer1**
>
> We thank for the review.

---

### Official Review · AnonReviewer4 · 2019-11-01
**Official Blind Review #4**

**Rating:** 3

**Review:**

The authors propose to combine planning methods like MCTS with an ensemble of value functions to a) estimate the value of leaf nodes of the search tree and b) use the ensemble estimate of uncertainty to guide exploration during MCTS search.
The MCTS rollouts are also used as optimization targets for the value function.

I believe this is a clear reject. On the one hand, the paper needs signficiantly more work on the writing and clarity. On the other hand I have several worries on the method and evaluation side.

Regarding the presentation of the paper:
Overall, the paper seems quite rushed. This is not a strong reason for rejection but should be improved in a future version. For example, punctuation and sentence structure is often wrong, the paper has only slighlty over 7 pages, a citation is undefined on p.7 and images and whitespace is formatted wrongly on occasion (e.g. top of page 6).
More importantly, on the content side, the experimental section is sufficiently clear and well written, however, the method description needs more detail and background information. The paper relies on several prior works which are referred to but not described (E.g. MCTS , the sampling mechanism by Osband et al. which they are using but not describing, the 'mask' from Osband et al which they are using but not describing).
Furthermore, the algorithm itself is not described in sufficient detail:
- How does the 'soft-penalization' work?
- How exactly does the mechanism "similar in fashing to" Thomson sampling work?
- Are you learning a model or do you have access to the true transition function?

Regarding the method:
I can't say anything definitive about the method as I'm not entirely clear how exactly it works. However, I have several worries that might need addressing:
- It seems to me that the method relies on access to the _true_ transition and reward function and not on a learned model. This is a big difference to much of the prior work they compare against. This also makes the comparison against any pure model free method like PPO much less meaningful.
- Similarly, manually avoiding dead-ends and loops is a very strong assumption
- Also, being able to distinguish and use a fixed ratio of "solved" and "unsolved" episodes is a strong assumption.
- The one main contribution seems to be a new way of how \phi_a(x) is defined. Their particular choice needs a clearer motivation. Furthermore, if there is more contribution and differences to prior work, highlighting them more would help the reader understand the contribution.
- As the work makes several strong assumptions regarding the environment and access to the model, significantly more work (e.g. ablation studies) is needed to clearly show which assumption and feature of the algorithm is important for performance (and ideally also why). For example (but that's just a first idea): To understand the impact of their choice of \phi vs. their planning architecture, it would be be interesting to maybe train PPO using an exploration bonus based on \phi. This would allow disentangling the contribution of: Access to the true model, "discrete-environment-tricks" like penalizing dead-ends, and exploration incentivication of \phi.

Edit:
Thank you for your response and the updated manuscript, which reads considerably better.
I also agree with your point regarding the strength of assumption regarding "solved" and "unsolved" episodes.

Consequently, I will raise my score to a "weak reject" to express that I think this is promising work.

I do believe that ablation studies would add a lot to the paper as they would allow one to see which of the (many) added components help how much, for example between the selection function $\phi$ and the various penalizations used.

**Experience Assessment:**

I have read many papers in this area.

**Review Assessment: Checking Correctness Of Derivations And Theory:**

I assessed the sensibility of the derivations and theory.

**Review Assessment: Checking Correctness Of Experiments:**

I assessed the sensibility of the experiments.

**Review Assessment: Thoroughness In Paper Reading:**

I read the paper at least twice and used my best judgement in assessing the paper.

---

> ### Author Response · Authors · 2019-11-15
> **Answer to AnonReviewer4**
>
> We thank the reviewer for a detailed review. We admit major deficiencies in the presentation of our work, which, we believe, improved significantly in the new uploaded version.
>
> Answering detailed comments:
> - the presentation of the method is rewritten. We hope that it is much clearer and addresses the reviewers concerns. In particular, we explain MCTS, mask, sampling mechanism either in Section 2 or Appendix.
> - As to the reviewer concerns regarding 'soft-penalization'. We penalize loops on two levels, mcts planner and the episodes. This is  now explained in Section 2 and Appendix A (the relevant parameters are penalty_p and penalty_e).
> - Regarding assumptions: In our work we assume access to the prefect model (realised by the simulator), which, hopefully clearly, is now stated in the introduction. Having a model enables to avoid loops (independently of the fact how the model is obtained).
> We admit that access to a model is a substantial assumption, however this is somewhat orthogonal to our main focus. In the future work, we would like to address learning models. There are a number of domains, like Sokoban, in which, we believe, learning of the model is much simpler than planning. In fact our very preliminary experiments indicate that learning model of Sokoban is is indeed likely to be doable.
> - We respectfully disagree with the statement that that having fixed ratio of “solved” and “unsolved” episodes is a strong assumption. For environments with a single (sparse) reward for completing a task (like the environments used in our experiments) it is a very natural concept. Furthermore, there is a growing body of literature concerning prioritised usage of “good” episodes, see e.g. [1,2,3].
> - Motivation behind particular choices of \phi_a are now presented in Section 2.
> - We admit that  having ablations would be nice to have and we will prepare them for the camera ready version. We reckon that this does not diminish the value of the method itself.
> - We like the idea of including the exploration bonus into model-free training and state it in the future work section.
>
> [1] Junhyuk Oh, Yijie Guo, Satinder Singh, and Honglak Lee.  Self-imitation learning. ICML 2018
> [2] Kaixiang Lin, Jiayu Zhou, Ranking Policy Gradient. arXiv:1906.09674, 2019
> [3] Yijie Guo, Jongwook Choi, Marcin Moczulski, Samy Bengio, Mohammad Norouzi, Honglak Lee, Efficient Exploration with Self-Imitation Learning via Trajectory-Conditioned Policy. arXiv:1907.10247 (2019)

---

### Official Review · AnonReviewer3 · 2019-11-04
**Official Blind Review #3**

**Rating:** 1

**Review:**

Uncertainty-Sensitive Learning and Planning with Ensembles
=====================================================

This paper investigates the use of uncertainty-aware estimates in solving planning problems (RL with access to simulator).
The proposed algorithm combines a learned model-free value estimate with MCTS planning.
An ensemble of neural networks is used to model posterior uncertainty in the value estimate and drive efficient exploration.


There are several things to like about this paper:
- The paper takes on several core issues in RL/planning research, most notably the synthesis of dealing with model-based and model-free uncertainty in RL.
- The general flavour of the paper + algorithm seems to be reasonable. The proposal to use ensemble uncertainty estimates to drive model-based MCTS is interesting, natural, and I think it's a good one.
- The proposed structure of the paper is quite nice, there is mostly a linear and logical progression of complexity in the experiments. This is nice to see clear benefits of the approach on the simplest possible settings and build up from there.
- The effort to open source code + implementation details is laudable.

However, there are several places where this paper falls short:
- In general, the claims and results of the paper are far too vague to be fully understood and replicated. Take the main algorithm 1, it really seems like more of a "sketch" of a very general family of algorithms, rather than a specific description of a clear algorithm.
- This vagueness is spread throughout the plots and figures as well... note that Figure 1 has no indication of how many steps have been evaluated, and Figure 2 has no indication for what value K > 0 was actually used. The clarity does not improve in Sections 3.2 and 3.3 where quite inconsistent performance metrics and presentations are presented.
- Generally, the writing could be tightened quite a lot. In particular I would encourage you to think about whether each statement you make is clearly supported by some theorem, experiment or plot in your paper. For example, on page 3 "We found this mechanism to be beneficial... see Section 3.3" but then it's not clear exactly what statement shows that particular part of the mechanism was helpful, versus other issues associated with ensemble learning. There are more than a few typos... the on(e) in Osband... akin to ??... might be obtained by choosing from (the) ensemble...
- It would be very helpful to clarify that the agent is given access to a simulator... so that this is not exactly the typical RL setting of sequential decision making. This should appear early in the paper.
- The code that is released with the paper is also quite confusing, it is not structured with a clear README and includes many sections of dead/commented code. I was hoping the code might rescue some of the clarity, but I think that still needs work.

Overall, I do think there is some interesting material here...
It's an important problem, and the core building blocks of combining model, value and uncertainty for better exploration is interesting.
However, I just think the actual paper is not clear enough on the details.
My belief is that going through this paper very methodically and carefully to make sure that every single detail + claim is rigorously supported would help this paper immensely.
For that reason I have to say that I think it's a "reject" in its current form.

**Experience Assessment:**

I have published in this field for several years.

**Review Assessment: Checking Correctness Of Derivations And Theory:**

I assessed the sensibility of the derivations and theory.

**Review Assessment: Checking Correctness Of Experiments:**

I assessed the sensibility of the experiments.

**Review Assessment: Thoroughness In Paper Reading:**

I read the paper at least twice and used my best judgement in assessing the paper.

---

> ### Author Response · Authors · 2019-11-15
> **Answer to AnonReviewer3**
>
> We thank for the review and comments. We admit various shortcomings especially in the text clarity. We have uploaded a new overhauled version of the paper, which we hope makes our work more accessible.
>
> Below we address particular concerns of the reviewer:
> - “main algorithm 1, it really seems like more of a sketch” - the description of the method has been rewritten. We provide pseudo-code for all the components of the method. This is done in Section 2 and Appendix A.
> - Figures descriptions, in particular Figures 1 and Figure 2, have been clarified.
> - We assume access to the perfect model (simulator so to speak), which is now, hopefully clearly, stated in the introduction. One could argue that in some cases, like Sokoban, the model is easy to learn, what is hard is planning. Having said that, learning models and using imperfect models is an exciting research direction.
> - As to the code, we have made some further clean-ups and improved the README file. Meanwhile, we have also been working on a completely new version of code, redesigned from scratch. We hope that it will also be of use for the community.
> - We made our best to clearly state our contribution as well as match our claims with the corresponding evidence.

---

### Public Comment · ~Anthony_Wittmer1 · 2019-09-28
**No code in provided github link even after 60 hours of submission deadline**

Hi,

No code is present in the repo of the github link. It is not fair to provide a placeholder link for code submissions (which impact the review process) and submit code taking considerable buffer time after submission deadline.

---

> ### Author Response · Authors · 2019-10-08
> **repo working**
>
> You're right. It took us somewhat longer to make a repo, which we are sure that is fully anonymous. Now it is up and running.

---

### Decision · Program_Chairs · 2019-12-19

**Decision:**

Reject

**Comment:**

The authors study planning problems with sparse rewards.
They propose a tree search algorithm together with an ensemble of value
functions to guide exploration in this setting.
The value predictions from the ensemble are combined in a risk sensitive way,
therefore biasing the search towards states with high uncertainty in value
prediction.
The approach is applied to several grid-world environments.

The reviewers mostly criticized the presentation of the material, in particular
that the paper provided insufficient details on the proposed
method. Furthermore, the comparison to model-free RL methods was deemed somewhat
lacking, as the proposed algorithm has access to the ground truth model.
The authors improved the manuscript in the rebuttal.

Based on the reviews and my own reading I think that the paper in it's current
form is below acceptance threshold. However, with further improved presentation
and baselines for the experiments, this has potential to be an important contribution.